Nutrient cycling characteristics along a chronosequence of forest primary succession in the Hailuogou Glacier retreat area, eastern Tibetan Plateau

Yang Danli 1 yangdanli5203@163.com
Luo Ji 2
Shu Shumiao 1
Hu Yan 1
Tang Hongsong 1
Li Xuemei 1
1 Neijiang Normal University , Neijiang , China
2 Institute of Mountain Hazards and Environment, Chinese Academy of Sciences , Chengdu , China
Sadeghi Seyed Mohammad Moein
Electronic publication date: 2025 Jan 31
Publication date: 2025
Volume: 13
Electronic Location ID: e18867
Received 2024 May 21; Accepted 2024 Dec 24
Copyright: © 2025 Yang et al.
Copyright year: 2025
Copyright holder: Yang et al.
License: This is an open access article distributed under the terms of the Creative Commons Attribution License, which permits unrestricted use, distribution, reproduction and adaptation in any medium and for any purpose provided that it is properly attributed. For attribution, the original author(s), title, publication source (PeerJ) and either DOI or URL of the article must be cited.
License URL: https://creativecommons.org/licenses/by/4.0/

Keywords: Primary succession, Nutrient allocation, Nutrient cycling, Glacier retreat area

Funding: Second Tibetan Plateau Scientific Expedition and Research Program (STEP) 2019QZKK0307 National Natural Science Foundation of China 41771062 This work was supported by the Second Tibetan Plateau Scientific Expedition and Research Program (STEP) (Grant No. 2019QZKK0307) and the National Natural Science Foundation of China (Grant No. 41771062). The funders had no role in study design, data collection and analysis, decision to publish, or preparation of the manuscript.

==============================
Background

The Hailuogou Glacier has been continuously retreating since the end of the Little Ice Age, resulting in a 125-year soil chronosequence and a complete primary forest succession sequence. Nutrient cycling and utilization are the foundation to forest succession processes and dynamic changes, directly influencing the structure and stability of ecosystems. However, our understandings on the characteristics of ecosystem nutrient accumulation and recycling during succession, especially in the context of primary succession within glacier retreat areas, remain limited. To address this, we investigated nutrient characteristics across six forest primary succession sites in the Hailuogou Glacier retreat area.

Methods

Six sites representing three forest stages: the pioneer plant stage (S1), the broad-leaved forest stage (S2–S4), and the coniferous forest stage (S5–S6). Three quadrats were established at each site, and measurements of biomass as well as soil characteristics were documented within each quadrat. Subsequently, we collected samples of vegetation, soil and litter. By measuring the concentrations of N, P, K, Ca, and Mg in vegetation and soil and combining with the data of the quadrat survey, the pools and nutrient characteristics of N, P, K, Ca, and Mg in various components of the ecosystem were calculated at each site.

Results

Our findings indicated that: (1) Nutrient pools, excluding the soil C layer, increased with forest primary succession, reaching 5,995.71 kg hm−2 N, 461.83 kg hm−2 P, 3,798.09 kg hm−2 K, 7,559.81 kg hm−2 Ca and 1,948.13 kg hm−2 Mg at site S6; however, the pools of P, K, and Mg in the Oa layer, and Ca and Mg in the tree layer, attained their peak levels at sites S3 to S4. (2) The pools of N, Ca, and Mg in the organic soil were significantly greater than vegetation. Although over 60% of the P and K were stored in the organic soil at site S1, these proportions shifted, with vegetation holding 60.71% of P and 56.86% of K at site S5. (3) Broad-leaved forests exhibited higher nutrient return, cycling, and absorption, thereby accelerating nutrient circulation and depleting soil nutrients to maintain growth. In contrast, coniferous forests were more efficient at nutrient utilization and storage, retaining nutrients and maintaining high biomass and productivity in nutrient-poor environments. Overall, these findings highlighted that the nutrients in each component of the ecosystem continue to accumulate with forest primary succession. Coniferous forests’ nutrient cycling mechanisms offer a competitive edge in nutrient-poor environments, enhancing ecosystem stability.

Introduction

The biogeochemical cycle, encompassing both the geochemical and biological processes, is essential to ecosystem development. The geochemical cycle includes the weathering of rocks, erosion, transportation, deposition, and subsequent lithification; while the biological cycle involves the absorption of by plants and their return to the soil through litter and woody debris. Microorganisms play a crucial role in converting these nutrients into forms that are advantageous for plants, thereby enhancing the formation and development of soil fertility (Schlesinger & Bernhardt, 2016; Soper et al., 2018; Johnson & Turner, 2019). Nutrient cycle, which includes the absorption, allocation, return, and release of nutrients within ecosystems, is a critical component of the biological cycle. The cycling and balance of elements significantly influence the level of net primary productivity, as well as the stability and sustainability of ecosystems, thereby shaping the overall structure of ecological systems (Switzer & Nelson, 1972; Reich & Oleksyn, 2004; Díaz et al., 2016).

The cycling of nutrients serves as a fundamental underpinning for the survival and development of ecosystems. Essential macronutrients such as nitrogen (N), phosphorus (P), potassium (K), calcium (Ca), and magnesium (Mg) are critical for plant growth, with plants absorbing these nutrients from the soil to sustain normal processes (Hagen-Thorn et al., 2006; Song et al., 2009; Zhang et al., 2020). Although their cycles exhibit corresponding spatial and temporal patterns of change (Melvin et al., 2015; Lambert & Turner, 2016; Jennifer & Erika, 2017), they show some convergence with vegetation succession. The cycling and utilization of nutrients significantly influence the structure and dynamics of communities, serving as a foundation for community succession processes. Conversely, the differences in tissue nutrient concentrations among species play a critical role in influencing changes in nutrient cycling patterns (Turner & Lambert, 2016). Thus, during early succession, soil nutrients are gradually transferred to forest ecosystems, resulting in a positive correlation between forest biomass and nutrient concentration (Shao et al., 2017). As forests mature, their reliance on soil for nutrients decreases, a pattern common in secondary forests (Powers & Marín-Spiotta, 2017; Zhou et al., 2016; Lambert & Turner, 2016; Paul et al., 2022). However, as forests develop, the continuous accumulation of both soil and forest nutrients, along with the mechanisms underlying nutrient cycling, may differ between primary and secondary forest succession.

Primary succession develops forest communities in previously non-vegetated areas, which are typically not affected by human or historical factors, can further our understanding of the dynamics of nutrients accumulation under natural conditions (Yang et al., 2021a; Kaneda, Angst & Frouz, 2020). Primary succession in glacial forelands is especially relevant due to rapid glacial retreat, which exposes new areas for plant colonization (Crocker & Major, 1995; Palo & Fornara, 2017; Glausen & Tanner, 2019). Therefore, the primary succession on glacial forelands provides a distinctive place to disentangle the accumulation and cycling characteristics of ecosystem nutrients effects on community successional trajectories. The Hailuogou Glacier, located on the southeastern edge of the Tibetan Plateau, has been continuously retreating since the end of the Little Ice Age, resulting in a continuous soil chronosequence of approximately 125 years and a complete vegetation succession sequence (Yang et al., 2014; Jiang et al., 2018; Wang et al., 2020). While many studies have explored biomass changes (Yang et al., 2015), soil phosphorus bioavailability (Zhou et al., 2013; Wu et al., 2014a), nutrient accumulation (Yang et al., 2014; He et al., 2019; Yang et al., 2021a) and stoichiometric characteristics (Jiang et al., 2018), the role of nutrient cycling in forest succession remains underexplored. In the Hailuogou Glacier retreat area, environmental factors like temperature and moisture vary minimally, making nutrient cycling a potential key driver of vegetation succession (Moen, Cairns & Lafon, 2008; Ducic, Milovanovic & Durdic, 2011). Thus, the relationship between nutrient cycling and primary succession will be further amplified and the patterns in nutrient cycling may reveal either causes or consequences of successional changes in plant communities here.

This study aims to elucidate the relationship between nutrient accumulation, cycling characteristics, and community succession in primary forest succession. We tested the following hypotheses: (1) As forest succession progresses, the pools of N, P, K, Ca, and Mg in soil and vegetation will progressively increase, but there will be differences in their distribution patterns. (2) The nutrient accumulation, cycling, and utilization are primary drivers of forest succession in the Hailuogou Glacier retreat area. To test these hypotheses, we established six continuous sampling sites along the primary succession sequence and analyzed the accumulation processes as well as the cycling and utilization characteristics of N, P, K, Ca, and Mg at different stages.

Materials and Methods

Study site

Gongga Mountain (29°20′–30°20′N, 101°30′–102°15′E) is located at the junction of the Sichuan Basin and the Qinghai-Tibet Plateau, in the transitional zone between the alpine zone of the Qinghai-Tibet Plateau and the warm and humid subtropical monsoon region. The Hailuogou Glacier retreat area (29°34′N, 102°59′E), located on the eastern slope of Gongga Mountain, belongs to the alpine temperate climate type, with an annual average temperature of 3.8 °C, more than 260 rainy days per year, and an annual average precipitation of approximately 1,960 mm. Since the Hailuogou Glacier began to retreat after the Little Ice Age, it has been less disturbed by humans, and the glacier retreat is significant without any glacial advance process. This has led to the formation of a complete forest primary succession sequence of approximately 2 km, accompanied by a continuous soil formation process (Bai et al., 2019).

The field surveys has been described in detail elsewhere (Yang et al., 2021b). Briefly, six sampling sites were selected for vegetation and soil samples collection at various glacier retreat time (Table 1). Specifically, Site 1 (S1) represents 15 y of glacial retreat and was invaded by pioneer herbs and trees, including Hippophae rhamnoides Linn., Salix spp., Populus purdomii Rehd., and several other leguminous herbs. Site 2 (S2), Site 3 (S3) and Site 4 (S4) represent glacier retreat after 35 y, 45 y and 57 y, respectively. During the broad-leaved forest stage (S2–S4), P.purdomii became the predominant species owing to its rapid growth rate and high photosynthetic rate, which meant that it was able to outcompete H. rhamnoides and Salix spp. Site 5 (S5) and Site 6 (S6) experienced glacial retreat after 85 y and 125 y, respectively. During the coniferous forest stage (S5–S6), P. purdomii was gradually replaced by Abies fabri (Mast.) Craib.

Table 1 Investigation of vegetation and soil at different glacier retreat years.

Sampling site	S1	S2	S3	S4	S5	S6	
Glacier retreat year	15 y	35 y	45 y	57 y	85 y	125 y	
Dominant plants	Astragalus adsurgens Pall., H. rhamnoides, Salix spp., P. purdomii	H. rhamnoides, Salix spp., P. purdomii	P. purdomii (half-mature), A. fabri	P. Purdomii (mature), A. fabri	P. purdomii, A. fabri	A. fabri	
Total living biomass (t hm−2)	7 (1)	121 (13)	199 (22)	225 (10)	291 (24)	366 (19)	
pH of organic soil	6.9 (0.3)	6.4 (0.5)	5.5 (0.5)	5.6 (0.4)	5.2 (0.4)	4.4 (0.4)	
Thickness (cm)	Oe layer	0.8 (0.1)	1.1 (0.2)	1.4 (0.2)	1.8 (0.3)	2.3 (0.9)	3.6 (0.8)	
Oa layer	—	1.8 (0.7)	2.6 (0.2)	3.8 (0.9)	4.6 (1.2)	5.4 (1.2)	
Bulk density (g cm−3)	Oe layer	0.13 (0.01)	0.12 (0.02)	0.11 (0.06)	0.12 (0.07)	0.19 (0.03)	0.35 (0.05)	
Oa layer	—	0.31 (0.11)	0.38 (0.10)	0.33 (0.09)	0.25 (0.06)	0.30 (0.09)	
Note:

Data shown as means with standard deviation in parentheses.

Vegetation sample collection

Our research is in the Gongga Mountain Nature Reserve, and the current study was carried out on the fixed sample plot set up by the research station. We work in close coordination with the management department of the Reserve. Our work is also part of the monitoring and managment of the Reserve. The general observation experiment did not need to be approved. Therefore, all field sampling mentioned in this study was permitted.

The sample collection method as previously described in Yang et al. (2021b). Specifically, we established three quadrats of 10 m × 10 m at each site, and recorded the species name, DBH, height and geographical coordinates of the tree with a diameter at breast height (DBH; 1.3 m height aboveground) >2 cm. The vegetation biomass estimated by harvesting the dominant tree species in each quadrat to obtain the species-specific allometric equations (Yang et al., 2015). Collection of vegetation samples (foliage, twig, stem, bark, and root) of all tree species at six sampling sites were conducted once in each year from 2015 through 2016 during the peak of deciduous foliage mass. Meanwhile, we established a quadrat of 5 m × 5 m to harvest the shrub biomass and a quadrat of 1 m × 1 m to harvest the herb and moss biomass in each 10 m × 10 m quadrat. The tree materials and understory vegetation materials were oven-dried at 60 °C to a constant weight. The dried samples were ground to a fine powder and used to measure the nutrient concentration. The N concentration was measured by an element analyzer (Vario Macro Cube C, Elementar, Wetzlar, Germany). The P, K, Ca and Mg concentrations were analyzed using an American Leeman Labs Profile Inductively Coupled Plasma-Atomic Emission Spectrometer (ICP-AES).

Litter and soil sample collection

According to the terrain, slope, and vegetation distribution, we established a 0.5 m × 0.5 m soil pit in each 10 m × 10 m quadrat. The sample collection method as previously described in Yang et al. (2021b). Specifically, we collected the litter (undecomposed) from the surface of the soil pit, and its biomass and nutrient concentration were determined in the same way as those of the vegetation material. For the soil sampling, no obvious A layer were found because of the short period of soil development, we established organic soil and C layer by using the protocol described elsewhere (Deyn, Raaijmakers & Putten, 2004). We used a steel auger drill (5 cm diameter) to obtain the soil-core sample in each pit. Then, all core samples were cut into horizontal slices of Oe (intermediate decomposed organic layer), Oa (highly decomposed organic layer) and C layer. We collected soil samples and measured their thickness, and a separate set of samples for soil bulk density was collected with a cutting ring in each soil layer. Whereafter, all soil samples were air dried at room temperature (–15 °C) by placing the samples on kraft paper and then measuring the moisture and calculating the bulk density. Each soil sample was ground in an agate mortar and sieved through a 200 mesh sieve. The N concentration of the soil was measured using the semimicro Kjeldahl method. The P, K, Ca and Mg concentrations were also analyzed as method used for the vegetation material.

Statistical analysis

The specific pool size of N, P, K, Ca, and Mg at each site are estimated based on the measured concentration of of N, P, K, Ca, and Mg in samples (tree, shrub, herb, and moss) and bulk soil and biomass densities:

(1) Poolveg=∑(Conveg×Biomassveg)

(2) Poolsoil=∑(Consoil×BulkDensitysoil×Depth×10−1).

The characteristics of the nutrient cycle were analyzed using biological cycle parameters such as retention, return, annual absorption, utilization coefficient, circulation coefficient and turnaround time. The retention is the annual accumulation or net accumulation of nutrient elements, calculated by multiplying the annual growth of the biomass of vegetation (Yang et al., 2015) by its nutrient concentration. The return was calculated by measuring the annual litter amount and nutrient concentration. Other biological cycle parameters as follows:

(3) Absorption=Retention+Return

(4) Utilizationcoefficient=Absorption/Totalvegetationpool

(5) Circulationcoefficient=Return/Absorption

(6) Absorptioncoefficient=Absorption/Totalorganicsoilpool

(7) Turnaroundtime=Totalvegetationpool/Return

where absorption is the total amount of nutrients absorbed by plants from the environment; utilization coefficient is used to reflect the magnitude of the storage rate of elements in the ecosystem, the greater the coefficient is, the lower the utilization efficiency of plants for elements; circulation coefficient reflects the residual amount of elements in the cycle process, the larger the coefficient is, the faster the element circulation rate and the higher the fluidity; absorption coefficient reflects the vegetation absorption of nutrients from the soil; the turnaround time reflects the time it takes for nutrients to go through a cycle (Pang et al., 2002; Yu et al., 2015).

Results

The element pools in the soil

The nutrient pools of N and P in the Oe layer gradually increased with soil development (Fig. 1). At site S6, the pools reached 1,395.34 kg hm−2 N, 124.37 kg hm−2 P, 1,469.73 kg hm−2 K, 3,349.68 kg hm−2 Ca, and 1,188.86 kg hm−2 Mg. In the Oa layer, the N and Ca pools gradually accumulated and reached maximum values of 3,040.94 kg hm−2 and 3,244.83 kg hm−2 at site S6, respectively; however, the K and Mg pools fluctuated with soil development and reached a maximum at site S3, and the P pool reached a maximum at site S4. The pools of N, P, K, Ca, and Mg in the C layer were the highest during the initial soil development stage and then decreased slightly afterwards. The pools of P, K, Ca, and Mg in all soil layers were greater in the C layer, while for N, the pool was greater in the organic soil. Additionally, in the organic soil, the pools of elements followed the order of Ca > N > K > Mg > P, while in the C layer, they followed the order of Ca > K > Mg > P > N.

Figure 1 N, P, K, Ca, and Mg pools in soil at different sample sites.

Lowercase letters indicate significant differences in the pools of N, P, K, Ca, and Mg in each soil layer at different sites (the data in the figure is mean + SD; the significance level of mean difference is 0.05).

The element pools in vegetation

Tree layer

The pools of N, P, and K in the tree layer gradually increased with forest primary succession, reaching 1,128.99 kg hm−2 N, 175.64 kg hm−2 P, and 1,315.61 kg hm−2 K at site S6 (Fig. 2). However, the pools of Ca and Mg reached the maximum 1,966.63 kg hm−2 and 175.91 kg hm−2 at site S4 , respectively. Based on the proportion of nutrient elements pools of dominant species in the tree layer at each sample site, N, P, K, and Mg were the most abundant in H. rhamnoides at site S1, accounting for 66.21% N, 59.43% P, 58.82% K and 43.83% Mg. From sites S2 to S4, the proportion of nutrient elements pools in P. purdomii were the highest, accounting for 86.11% N, 88.06% P, 92.48% K, 95.96% Ca, and 92.97% Mg at site S4. While at sites S5 and S6, the proportion of nutrient elements stored in A. fabri was significantly higher than that in P. purdomii, and almost all A. fabri occupied at site S6.

Figure 2 N, P, K, Ca, and Mg pools in the tree layer at the different sampling sites.

Understory vegetation

The pools of N, P, K, Ca, and Mg in the understory vegetation gradually increased with forest primary succession, reaching 429.75 kg hm−2 N, 31.94 kg hm−2 P, 204.46 kg hm−2 K, 282.79 kg hm−2 Ca, and 42.02 kg hm−2 Mg at site S6, and the order of the pools were N > Ca > K > Mg > P (Fig. 3). At site S1, the proportions of nutrient elements pools in the herb were relatively high, accounting for 74.15% N, 59.06% P, 79.16% K, 55.47% Ca and 48.62% Mg. At site S2, the proportions of P, Ca and Mg were highest in the moss, accounting for 50.02% P, 52.30% Ca, and 54.82% Mg. From sites S3 to S6, most of the nutrients in understory vegetation were mainly stored in shrub, accounting for 74.72% N, 70.02% P, 72.12% K, 73.07% Ca and 60.62% Mg at site S6. It was observed that the proportions of these elements in the shrub were less than 5% during the early stage of this succession sequence. However, with the increase in shrub biomass (Yang et al., 2015), particularly at site S6, the proportions of the N, P, K, Ca, and Mg pools within the shrub exceeded 50 %. This finding indicated that as the understory vegetation continuously developed, the shrub became the primary storage units for nutrient elements in the understory vegetation.

Figure 3 N, P, K, Ca, and Mg pools in understory vegetation at different sample sites.

Proportion of the element pool in the vegetation and organic soil

Due to the extremely low N concentration in the C layer and its primary function in nutrient cycling within organic soil, this study focused exclusively on the organic soil (Oe and Oa layers) and vegetation when addressing element distribution within the ecosystem (Fig. 4). As primary succession progresses, notable differences in the distribution of N, P, K, Ca, and Mg were observed between organic soil and vegetation. These differences were primarily manifested as follows: (1) At site S1, the limited duration of soil development and the low N concentration (only 0.44 g kg−1) resulted in only 4.64% of N being stored in organic soil. In contrast, at site S6, a significant increase was observed, with 74.03% of N being retained in organic soil, while vegetation contributed merely 25.97%. Consequently, soil has emerged as the predominant N pool within the ecosystem. (2) More than 60% of the P and K were predominantly stored in organic soil at site S1, with a relatively small proportion found in vegetation. However, as the pools of P and K within the vegetation continuously increased, their proportions surpassed those in the organic soil, reaching 60.71% for P and 56.86% for K at site S5. We proposed that a substantial amount of soil P and K has been transformed into biomass P and K pools. (3) At each site, the proportions of Ca and Mg pools in organic soil exceeded those observed in vegetation, establishing organic soil as the primary storage unit for Ca and Mg within the ecosystem.

Figure 4 Proportion of the element pool in the vegetation and organic soil.

The capital letters indicate significant differences in the proportions of N, P, K, Ca, and Mg in the organic soil at the different sites; the lowercase letters indicate significant differences in the proportions of N, P, K, Ca, and Mg in the total vegetation at the different sites (the data in the figure is mean + SD; the significance level of mean difference is 0.05).

The cycling characteristics of N, P, K, Ca, and Mg

The accumulation of N, P, and K in both vegetation and organic soil were observed to reach their peak at site S6 (Figs. 5–9). However, the accumulation of Ca and Mg reached its peak at site S4, whereas their accumulation in organic soil was highest at sites S6 and S1, respectively. Furthermore, the retention and absorption of N and K in vegetation were found to reach their maxima at site S6. In contrast, the retention and absorption of P, Ca, and Mg peaked at site S4. In addition, the absorption coefficient exhibited a gradual decrease throughout the process of forest succession. The annual returns of N and P were highest at site S4, whereas the maximum returns for K, Ca, and Mg occurred at site S6. Based on the utilization characteristics of elements, a lower utilization coefficient signifies a higher efficiency of element use by vegetation. The utilization coefficients of N, P, K, and Mg were found to be lowest at site S6. In contrast, the utilization coefficient of Ca exhibited its lowest value at site S4. A higher cycling coefficient of elements suggests a more rapid cycling rate of the element in vegetation. The cycling coefficients of N and K reached their highest levels at site S3. Conversely, P exhibited the highest coefficient at site S2, Ca was maximal at site S1, and Mg peaked at site S6. In addition, the turnaround times of N and K were observed to be the longest at site S5, P exhibited the longest turnaround time at site S6, while Ca and Mg showed their longest turnaround times at site S4.

Figure 5 The characteristics of N cycling and utilization of elements at different sample sites.

Figure 6 The characteristics of P cycling and utilization of elements at different sample sites.

Figure 7 The characteristics of K cycling and utilization of elements at different sample sites.

Figure 8 The characteristics of Ca cycling and utilization of elements at different sample sites.

Figure 9 The characteristics of Mg cycling and utilization of elements at different sample sites.

Discussion

Element dynamics in this succession sequence

At the early stage of this succession sequence (S1), N was primarily stored in vegetation. Several studies indicate that biological N fixation plays a crucial role in ecosystem N supply (Clevaland et al., 2022; Liang et al., 2022). The N concentration in the fine sand of Hailuogou Glacier sediment is extremely low, around 0.05 g kg−1 (Yang et al., 2021b). But the initial N-fixing colonizers serve to facilitate the establishment of late-successional dominants and promote soil N accumulation (Chapin et al., 1994; Menge & Hedin, 2009; Pérez et al., 2016), such as A.adsurgens and H.rhamnoides in this succession sequence (Sun et al., 2022). The N fixation rates for the roots of leguminous plants, such as A.adsurgens, ranged from 0.15 to 6.21 mg N g−1 d−1, while the roots of elaegnaseae species like H.rhamnoides exhibited rates between 0.13 and 5.36 mg N g−1 d−1. Notably, during the early stages of this forest succession sequence, the N fixation rate from root nodules accounted for approximately 94.8% of the overall ecosystem N fixation rate (Zhang, 2020). Thus, the N storage in the herb layer constituted 74% of that found in the understory layer, while the H.rhamnoides accounted for 66% of the N storage in the tree layer at site S1. Additionally, atmospheric N deposition serves as a significant source of N within ecosystems. The total rate of atmospheric N deposition in the study area was approximately 8 kg hm−2 y (Chang et al., 2018). Nevertheless, previous studies have shown that the N accumulation rate in organic soils (41 kg hm−2 y) exceeds the atmospheric N deposition rate (Yang et al., 2021b). Thus, the rate of N deposition was insufficient to account for the substantial increase in organic soil N. Consequently, biological N fixation may one important source of N accumulation within ecosystems during this succession sequence. Notably, over 125 years, organic soil stored more than twice the amount of N as vegetation, indicating soil as the primary N reservoir.

The P and K pools also increased with succession. Initially, the organic soil held larger P and K pools than vegetation at S1, but by S6, vegetation pools were equivalent to those in the soil. Wu reported that the bioavailable P in the soil increased significantly after 30 years of glacier retreat within this sequence. Moreover, the development of A. fabri in this context led to a decrease in soil pH, with values declining from 8.5 at the end of glaciation to 4.4 at site S6 (Wu et al., 2014b). This process can enhance the dissolution of certain silicate minerals, subsequently facilitating the production of bioavailable phosphorus P to satisfy the nutritional requirements of various types of vegetation (Zhou et al., 2013). The antagonism between Ca and K can impede K absorption when Ca levels are high, but as Ca in the soil steadily declines with succession, plants absorb more K from the soil. Consequently, K gradually accumulates in vegetation, surpassing soil levels at later stages. All of these studies corroborate our finding that a significant portion of the organic soil P and K at older sites has been converted into biomass P and K pools. Particularly after 85 years of recession, the P and K pools in vegetation exceed those in the organic soil, especially in coniferous-dominated forests. Furthermore,Ca and Mg pools in organic soil remained higher than those in vegetation throughout the succession, indicating that soil is the primary storage reservoir for Ca and Mg.

In the understory vegetation, the accumulation of elements occurred most rapidly during the broad-leaved forest stage (sites S4–S5). This phenomenon can be primarily attributed to the broad-leaved forest stage, during which P. purdomii emerged as the dominant species within the tree layer. The expansion of its ecological niche resulted in heightened intra- and interspecific competition, thereby reinforcing self-thinning and other thinning effects within the population (Yang et al., 2015). The emergence of numerous forest gaps within the community provides favorable conditions for the growth of shrubs and hers plants beneath the trees. This phenomenon leads to a significant increase in both the diversity and abundance of these plant species in that specific environment (Yang et al., 2021a). This phenomenon also resulted in an accelerated accumulation of nutrients within the understory vegetation during this period. The nutrient accumulation in understory vegetation is intricately linked to habitat changes prompted by the succession of dominant species.

The mechanism of element cycling in forest primary succession

Nutrient cycling and utilization mechanisms

Coniferous trees display higher nutrient utilization efficiency and lower cycling rates, enabling them to thrive in nutrient-poor environments during primary succession. At site S1, the duration of soil development was relatively brief, and the concentration of N in the organic soil measured only 0.44 g kg−1. Pioneer plant species, primarily A. adsurgens and H. rhamnoides, demonstrated a superior ability to adapt to nutrient-poor habitats under conditions of limited fertility by fixing atmospheric N through root nodules. This process enhanced the nutrient composition of the soil. However, at site S1, the low biomass and litterfall in the tree layer led to relatively rapid cycling rates of N, P, and K. Nonetheless, the utilization rate was found to be low, resulting in insignificant nutrient cycling. During the broad-leaved forest stage, specifically from sites S2 to S4, which were predominantly occupied by P. purdomii, the cycling rates of N, P, and K were comparatively elevated. This observation indicated that broad-leaved tree species were able to acquire nutrients through a more accelerated nutrient cycling process, characterized by shorter nutrient cycling periods. This facilitated their adaptation to relatively nutrient-deficient environments during the early stages of primary succession. However, a higher intensity of nutrient cycling in trees correlates with an increased consumption of accumulated elements from the soil. This phenomenon is not conducive to the accumulation of soil nutrients (Tian, Xiang & Yan, 2004; Luo et al., 2005). At sites S5 and S6, a coniferous forest stage dominated by A. fabri was established. The coniferous forest demonstrated higher nutrient utilization efficiency and a lower nutrient cycling rate, alongside enhanced nutrient retention capabilities. These characteristics contribute to a reduction in soil nutrient loss, thereby facilitating the maintenance of elevated levels of stand biomass and productivity. The nutrient cycling strategy of coniferous forests offers a competitive advantage in nutrient-poor environments, contributing to ecological stability.

Additionally, it is important to highlight that Ca exhibits contrasting recycling characteristics compared to N, P, and K across different forest stages. During the coniferous forest stage, Ca cycling was rapid with short turnover times but low utilization efficiency. In contrast, during the broad-leaved forest stage, slower Ca cycling and longer turnover times, resulted in higher utilization efficiency, leading to greater Ca accumulation in vegetation. Furthermore, the turnover times of elements followed this order: K > P > Ca > Mg > N in broad-leaved forests and K > P > Mg > N > Ca in coniferous forests. These patterns suggest that N and Ca are highly active, while K is least active throughout the succession process in the Hailuogou Glacier retreat area.

Compare the cycling and utilization characteristics of nutrients

We conducted a comparative analysis of the vertical zonal vegetation on Gongga Mountain and the vegetation corresponding to the successional stages in the Hailuogou Glacier retreat area (Table 2). We observed that in the broad-leaved forest, the utilization and circulation coefficients of N, P, and K in the Hailuogou Glacier retreat area were lower than those found in the vertical zone (except the utilization coefficient of P). Similarly, in coniferous forests, the coefficients in the Hailuogou Glacier retreat area were also lower than those in the vertical zone (except the cycling coefficient of N). This slower nutrient cycling and higher utilization efficiency indicate a more conservative nutrient utilization strategy during primary succession, promoting nutrient storage and minimizing loss. This approach enables the ecosystem to reach a climax community within 125 years, consistent with vertical zonation patterns.

Table 2 Comparing the nutrient cycling coefficient of the same forest type in Gongga Mountain vertical zone and Hailuogou Glacier retreated area.

Forest type	Area	Utilization coefficient	Cycling coefficient	
N	P	K	N	P	K	
Broad-leaved forest	vertical zone	0.103	0.083	0.056	0.547	0.453	0.318	
Hailuogou Glacier retreated area	0.086	0.117	0.053	0.487	0.170	0.157	
Coniferous forest	vertical zone	0.072	0.049	0.044	0.288	0.281	0.184	
Hailuogou Glacier retreated area	0.062	0.041	0.041	0.334	0.164	0.167	
Note:

The coefficients of broad-leaved forest in Hailuogou Glacier retreated area were calculated as the mean values of site S2–S4; the coefficients of coniferous forest were selected at site S6; the coefficients of vertical zone from Luo et al. (2005).

In addition, we further conducted a comparative analysis of the nutrient characteristics across various coniferous forests in the eastern Tibetan Plateau (Table 3). We observed that the utilization coefficients for N, P, K, Ca and Mg in Spruce and Abies forests (the dominant species at site S6 is A. fabri) were similar, but the circulation coefficients were notably higher in Abies forest. This difference may be due to elevation, the Spruce forests are located at an elevation of 3,200 m, while the Abies forests are found at a lower altitude of 2,900 m. It can be observed that the variation in altitude affects temperature, which in turn influences soil microorganisms. The activity of these microorganisms increases with rising temperatures, facilitating the decomposition of organic matter and promoting nutrient cycling within the soil (Jansson & Hofmockel, 2020; Qu et al., 2023). Therefore, a lower elevation of coniferous forests correlates with a faster nutrient cycling rate. The nutrient cycling rates of coniferous forests may exhibit heightened sensitivity to variations in temperature. However, temperature appears to have minimal influence on the nutrient utilization within Spruce and Abies forests, as the nutrient utilization coefficients across different coniferous forest types are consistently similar.

Table 3 Comparing the nutrient utilization and cycling coefficients of the natural coniferous forest.

Forest type	Area	Utilization coefficient	Cycling coefficient	
N	P	K	Ca	Mg	N	P	K	Ca	Mg	
Spruce forest	Alpine area on the eastern Tibetan Plateau	0.06	0.04	0.05	0.06	0.06	0.34	0.11	0.16	0.32	0.31	
Abies forest	Hailuogou Glacier retreated area	0.06	0.04	0.04	0.07	0.06	0.33	0.16	0.17	0.54	0.46	
Note:

The coefficients of Abies forest were selected at site S6, and the coefficients of Spruce forest from Pang et al. (2002).

Conclusions

Global climate change is anticipated to lead to a heightened occurrence of glacial retreat, underscoring the critical need for an in-depth understanding of nutrient dynamics in areas undergoing primary succession. With the primary succession of forests in the Hailuogou Glacier retreat area, the pools of N, P, K, Ca, and Mg increased in both organic soil and vegetation. The observed results can be attributed primarily to the accumulation of elements stemming from soil development, as well as an increase in vegetation biomass. There were significant differences in the distributions of N, P, K, Ca, and Mg between organic soil and vegetation. In particular, the N pool in organic soil showed a marked increase, establishing it as the primary N reservoir within the ecosystem. Conversely, the proportions of P and K pools in organic soil decreased throughout the ecosystem, suggesting that a portion of these nutrients was transferred to vegetation. Ca and Mg primarily accumulated in organic soil, being present in relatively smaller proportions within vegetative tissues.

In terms of the cycling characteristics of N, P, K, Ca, and Mg, during the early stages of succession, the limited biomass and litterfall from vegetation lead to the development of only an Oe layer in the organic soil. As a result, biogeochemical cycling characteristics were not significantly pronounced. With the succession of forests, the coefficients for nutrient return, nutrient cycling, and nutrient absorption were higher during the broad-leaved forest stage. This suggested that broad-leaved forests maintain growth by accelerating nutrient circulation while depleting soil nutrients. In contrast, the efficiency of nutrient utilization and nutrient storage were greater at the coniferous forest stage, indicating that coniferous forests were more effective in retaining nutrients within plant biomass and minimizing soil nutrient loss. Therefore, the capacity of coniferous forests plays a crucial role in maintaining high levels of forest biomass and productivity. Furthermore, the nutrient recycling mechanisms inherent to coniferous forests enhance their competitive advantage over other species, ultimately contributing to the establishment of a climax community. Overall, these findings highlighted that the nutrients within each component of the ecosystem continue to accumulate during forest primary succession. The nutrient cycling mechanisms in coniferous forests facilitate their competitive advantage over other species in nutrient-poor environments and contribute to their stability. Understanding the characteristics of nutrient cycling throughout primary succession is essential for enhancing the recovery of degraded ecosystems and for predicting forest migration patterns in response to future climate change. We recommend further studies to systematically explore the accumulation and cycling of nutrient elements in vegetation succession ecosystems across other deglaciated regions.

Supplemental Information

Supplemental Information 1 Biomass of vegetation in each sampling site of primary succession.

Supplemental Information 2 Elemental storage of soil and vegetation in each sampling site of primary succession.

We thank Xun Wang for his help in sampling normal analysis. We thank Wei Li for his valuable suggestions in the process of revising the manuscript. We gratefully acknowledge the anonymous reviewers for their comments, which significantly improved the manuscript.

Additional Information and Declarations

Competing Interests

The authors declare that they have no competing interests.

Author Contributions

Danli Yang conceived and designed the experiments, performed the experiments, analyzed the data, prepared figures and/or tables, authored or reviewed drafts of the article, and approved the final draft.

Ji Luo conceived and designed the experiments, performed the experiments, made other contributions, and approved the final draft.

Shumiao Shu analyzed the data, prepared figures and/or tables, and approved the final draft.

Yan Hu analyzed the data, authored or reviewed drafts of the article, and approved the final draft.

Hongsong Tang performed the experiments, authored or reviewed drafts of the article, and approved the final draft.

Xuemei Li performed the experiments, prepared figures and/or tables, and approved the final draft.

Field Study Permissions

The following information was supplied relating to field study approvals (i.e., approving body and any reference numbers):

Our research is in the Gongga Mountain Nature Reserve, and the current study was carried out on the fixed sample plot set up by the research station. We work in close coordination with the management department of the Reserve. Our work is also part of the monitoring and managment of the Reserve. The general observation experiment did not need to be approved. Therefore, all field sampling mentioned in this study was permitted.

Data Availability

The following information was supplied regarding data availability:

The raw measurements are available in the Supplemental Files.

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
