# Peer review of "Nutrient cycling characteristics along a chronosequence of forest primary succession in the Hailuogou Glacier retreat area, eastern Tibetan Plateau"

_PeerJ, doi:10.7717/peerj.18867_

## Round 0.1 · original submission · Minor Revisions

Dear authors,

I hope this message finds you well. Thank you for submitting your manuscript titled "Nutrient Cycling Characteristics along a Chronosequence of Forest Primary Succession in the Hailuogou Glacier Retreat Area, Eastern Tibetan Plateau" to our journal. The manuscript has been reviewed by three experts, and we have received valuable feedback aimed at strengthening your work.

I have outlined the minor revisions required before the manuscript can be considered for publication.

Cheers,
Seyed Mohammad Moein Sadeghi (Editor)

Reviewer 1 ·

Basic reporting

Some sections, particularly in the "Materials & Methods" and "Results" sections, contain lengthy, complex sentences that can be difficult to follow. Simplify and break down long sentences into shorter, more concise statements. Ensure each paragraph focuses on a single idea. While the introduction provides some context, it lacks a clear statement of the research gap and the significance of the study. It should better articulate the importance of the study and how it contributes to the field of nutrient cycling and forest succession. Some methodological details are insufficiently described, such as the specific techniques used for sample collection and analysis. Figures and tables are relevant but need clearer labeling and more descriptive captions. All visual data should be easily interpretable without referring back to the text. The literature cited is relevant and generally well-integrated into the manuscript. However, there could be a more comprehensive comparison of the study's findings with existing literature, especially in the discussion section.

Experimental design

While the methods are generally well-described, some aspects, such as the specific statistical methods used for data analysis, are not detailed enough. This lack of detail can hinder the replication of the study. Provide more comprehensive descriptions of the statistical analyses, including the software used, specific tests applied, and criteria for significance. The study does not include longitudinal data collected over multiple years, which could provide deeper insights into temporal variations in nutrient cycling. The methods section provides detailed descriptions of sample collection and analysis techniques for soil and vegetation. This includes specific information on the tools and procedures used, enhancing the reproducibility of the study. The discussion section provides a broad overview of the findings but lacks depth in interpreting the implications of the results. The discussion could be more thorough in explaining the significance of the findings in relation to existing literature. Deepen the analysis by comparing and contrasting the study’s results with previous research. Discuss the broader ecological implications and potential mechanisms underlying the observed nutrient cycling patterns.

Validity of the findings

The paper does not explicitly assess the impact and novelty of its findings within the broader context of ecological research. While the study presents original data and insights into nutrient cycling during forest succession in a glacial retreat area, it could benefit from a more explicit discussion of how these findings advance current knowledge and their potential implications for future research. you can include a dedicated section that explicitly discusses the impact and novelty of the study’s findings. Highlight how the research contributes new insights to the field and its potential implications for ecological theory and practice. Provide more detailed information on the statistical analyses performed, including the software used, specific tests applied, and criteria for significance. This transparency will enhance the credibility and reproducibility of the findings.

Additional comments

Overall, the paper presents a valuable and original contribution to the understanding of nutrient cycling in forest primary succession within a glacial retreat area. The study is well within the aims and scope of the journal, with a clear and well-defined research question. the discussion does not consider alternative explanations or potential limitations of the study, such as sampling biases or the constraints of the chrono sequence approach. There is also a tendency to overgeneralize the findings, which may not be fully supported by the data, especially concerning their applicability to other glacial retreat areas. The paper would benefit from a more detailed discussion of methodological limitations and a more cautious approach to generalizing results. Furthermore, the absence of explicit future research directions weakens the potential for guiding subsequent studies. The statistical analysis section needs more detailed explanations of the methods used and their appropriateness, enhancing the transparency and reproducibility of the findings. Addressing these weaknesses through a more thorough and critical discussion, better integration with existing research, and detailed methodological transparency would significantly improve the paper's quality and contribution to the field.

Reviewer 2 ·

Basic reporting

Expand on the Importance of the Study:
The introduction provides a good overview of nutrient cycling and succession, but the significance of studying these processes in the Hailuogou Glacier retreat area could be emphasized more. Why is this study important in the broader context of ecological research? How does it contribute to our understanding of primary succession or nutrient cycling?

Address Potential Gaps in Previous Research:
It would be beneficial to briefly highlight any gaps or limitations in existing research on nutrient cycling during primary succession, which your study aims to address. This will help position your research within the broader scientific context.

Reference Formatting:
Ensure that the references are consistently formatted, particularly with spacing and punctuation. For instance, there should be a space between the author names, so in line 41, "Schlesinger et al., 2016" must be used instead of "Schlesinger et al.,2016".

Sentence Structure and Grammar:
There are a few long sentences that could be broken down for clarity. For example, lines 47 to 50, "The cycling of nutrients is a crucial foundation for the survival and development of ecosystems..." could be split to avoid overloading the reader with too much information at once. Simplifying complex sentences can make the text more accessible.

Experimental design

Study Site Description:
The description of the study site is clear, but it could be enhanced by providing additional context or relevance. For example, briefly mention why the specific conditions of the Hailuogou Glacier retreat area make it an ideal location for this study. Additionally, provide some context on the significance of the 2 km forest primary succession sequence.

Grammar and Language:
Ensure grammatical consistency throughout. For example, in line 113, in "and recorded the the species name," the word "the" is repeated.

Validity of the findings

no comment

Additional comments

no comment

Reviewer 3 ·

Basic reporting

References must be reviewed, mainly those remarked, in the text, by red bold (references "Jonh & Mary" cited by "John et al.").

Figure 5 is extremely difficult to understand: too much information in a single image.

Most of suggestions are given along the text.

Experimental design

Is likeable to add a new figure with study site map / location.
Also, a better description of soil types, and vegetation composition in all treatments (specially when a huge discussion of leguminous effects are presented).

Validity of the findings

Nothing to add (see the text)

Additional comments

No further comments

Annotated reviews are not available for download in order to protect the identity of reviewers who chose to remain anonymous.

---

## Round 0.2 · accepted · Accept

I am happy with the revised version of your manuscript, and congratulations!

Reviewer 1 ·

Basic reporting

After reviewing the revised manuscript, I find that the authors have addressed the comments satisfactorily. The revised version has adequately addressed all the concerns raised during the initial review. I have no further comments, and I am satisfied with the improvements made and have no objections to its acceptance.

Experimental design

After reviewing the revised manuscript, I find that the authors have addressed the comments satisfactorily. The revised version has adequately addressed all the concerns raised during the initial review. I have no further comments, and I am satisfied with the improvements made and have no objections to its acceptance.

Validity of the findings

After reviewing the revised manuscript, I find that the authors have addressed the comments satisfactorily. The revised version has adequately addressed all the concerns raised during the initial review. I have no further comments, and I am satisfied with the improvements made and have no objections to its acceptance.

Additional comments

After reviewing the revised manuscript, I find that the authors have addressed the comments satisfactorily. The revised version has adequately addressed all the concerns raised during the initial review. I have no further comments, and I am satisfied with the improvements made and have no objections to its acceptance.

Reviewer 2 ·

Basic reporting

All previous comments were addressed.

Experimental design

All previous comments were addressed.

Validity of the findings

All previous comments were addressed.